# IS ATTENTION ALL THAT NERF NEEDS?

**Mukund Varma T[1]\*, Peihao Wang[2]\*, Xuxi Chen[2], Tianlong Chen[2],**
**Subhashini Venugopalan[3], Zhangyang Wang[2]**

[1]Indian Institute of Technology Madras, [2]University of Texas at Austin, [3]Google Research
`mukundvarmat@gmail.com, vsubhashini@google.com`
`{peihaowang,xxchen,tianlong.chen,atlaswang}@utexas.edu`

## ABSTRACT

We present *Generalizable NeRF Transformer* (**GNT**), a transformer-based architecture that reconstructs Neural Radiance Fields (NeRFs) and learns to render novel views on the fly from source views. While prior works on NeRFs optimize a scene representation by inverting a handcrafted rendering equation, GNT achieves neural representation and rendering that generalizes across scenes using transformers at two stages. (1) The *view transformer* leverages multi-view geometry as an inductive bias for attention-based scene representation, and predicts coordinate-aligned features by aggregating information from epipolar lines on the neighboring views. (2) The *ray transformer* renders novel views using attention to decode the features from the view transformer along the sampled points during ray marching. Our experiments demonstrate that when optimized on a single scene, GNT can successfully reconstruct NeRF without an explicit rendering formula due to the learned ray renderer. When trained on multiple scenes, GNT consistently achieves state-of-the-art performance when transferring to unseen scenes and outperform all other methods by ~10% on average. Our analysis of the learned attention maps to infer depth and occlusion indicate that *attention* enables learning a physically-grounded rendering. Our results show the promise of transformers as a universal modeling tool for graphics. Please refer to our project page for video results: `https://vita-group.github.io/GNT/`

## 1 INTRODUCTION

Neural Radiance Field (NeRF) (Mildenhall et al., 2020) and its follow-up works (Barron et al., 2021; Zhang et al., 2020; Chen et al., 2022) have achieved remarkable success on novel view synthesis, generating photo-realistic, high-resolution, and view-consistent scenes. Two key ingredients in NeRF are, (1) a coordinate-based neural network that maps each spatial position to its corresponding color and density, and (2) a differentiable volumetric rendering pipeline that composes the color and density of points along each ray cast from the image plane to generate the target pixel color. Optimizing a NeRF can be regarded as an inverse imaging problem that fits a neural network to satisfy the observed views. Such training leads to a major limitation of NeRF, making it a time-consuming optimization process for each scene (Chen et al., 2021a; Wang et al., 2021b; Yu et al., 2021).

Recent works Neuray (Liu et al., 2022), IBRNet (Wang et al., 2021b), and PixelNerf (Yu et al., 2021) go beyond the coordinate-based network and rethink novel view synthesis as a cross-view image-based interpolation problem. Unlike the vanilla NeRF that tediously fits each scene, these methods synthesize a generalizable 3D representation by aggregating image features extracted from seen views according to the camera and geometry priors. However, despite showing large performance gains, they unexceptionally decode the feature volume to a radiance field, and rely on classical volume rendering (Max, 1995; Levoy, 1988) to generate images. Note that the volume rendering equation adopted in NeRF over-simplifies the optical modeling of solid surface (Yariv et al., 2021; Wang et al., 2021a), reflectance (Chen et al., 2021c; Verbin et al., 2021; Chen et al., 2022), inter-surface scattering and other effects. This implies that radiance fields along with volume rendering in NeRF are not a universal imaging model, which may have limited the generalization ability of NeRFs as well.

---

*Equal contribution.

In this paper, we first consider the problem of transferable novel view synthesis as a two-stage information aggregation process: the multi-view image feature fusion, followed by the sampling-based rendering integration. Our key contributions come from using transformers (Vaswani et al., 2017) for both these stages. Transformers have had resounding success in language modeling (Devlin et al., 2018) and computer vision (Dosovitskiy et al., 2020) and their "self-attention" mechanism can be thought of as a universal trainable aggregation function. In our case, for volumetric scene representation, we train a *view transformer*, to aggregate pixel-aligned image features (Saito et al., 2019) from corresponding epipolar lines to predict coordinate-wise features. For rendering a novel view, we develop a *ray transformer* that composes the coordinate-wise point features along a traced ray via the attention mechanism. These two form the *Generalizable NeRF Transformer* (**GNT**).

GNT simultaneously learns to represent scenes from source view images and to perform scene-adaptive ray-based rendering using the learned attention mechanism. Remarkably, GNT predicts novel views using the captured images without fitting per scene. Our promising results endorse that transformers are strong, scalable, and versatile learning backbones for graphical rendering (Tewari et al., 2020). Our key contributions are:

1. A *view transformer* to aggregate multi-view image features complying with epipolar geometry and to infer coordinate-aligned features.
2. A *ray transformer* for a learned ray-based rendering to predict target color.
3. Experiments to demonstrate that GNT's fully transformer-based architecture achieves state-of-the-art results on complex scenes and cross-scene generalization.
4. Analysis of the attention module showing that GNT learns to be depth and occlusion aware.

Overall, our combined *Generalizable NeRF Transformer* (GNT) demonstrates that many of the inductive biases that were thought necessary for view synthesis (e.g. persistent 3D model, hard-coded rendering equation) can be replaced with attention/transformer mechanisms.

## 2    RELATED WORK

**Transformers** (Vaswani et al., 2017) have emerged as a ubiquitous learning backbone that captures long-range correlation for sequential data. It has shown remarkable success in language understanding (Devlin et al., 2018; Dai et al., 2019; Brown et al., 2020), computer vision (Dosovitskiy et al., 2020; Liu et al., 2021), speech (Gulati et al., 2020) and even protein structure (Jumper et al., 2021) amongst others. In computer vision, (Dosovitskiy et al., 2020) were successful in demonstrating Vision Transformers (ViT) for image classification. Subsequent works extended ViT to other vision tasks, including object detection (Carion et al., 2020), segmentation (Chen et al., 2021b; Wang et al., 2021c), video processing (Zhou et al., 2018a; Arnab et al., 2021), and 3D instance processing (Guo et al., 2021; Lin et al., 2021). In this work, we apply transformers for view synthesis by learning to reconstruct neural radiance fields and render novel views.

**Neural Radiance Fields** (NeRF) introduced by Mildenhall et al. (2020) synthesizes consistent and photorealistic novel views by fitting each scene as a continuous 5D radiance field parameterized by an MLP. Since then, several works have improved NeRFs further. For example, Mip-NeRF Barron et al. (2021; 2022) efficiently addresses scale of objects in unbounded scenes, Nex (Wizadwongsa et al., 2021) models large view dependent effects, others (Oechsle et al., 2021; Yariv et al., 2021; Wang et al., 2021a) improve the surface representation, extend to dynamic scenes (Park et al., 2021a;b; Pumarola et al., 2021) , introduce lighting and reflection modeling  (Chen et al., 2021c; Verbin et al., 2021), or leverage depth to regress from few views (Xu et al., 2022; Deng et al., 2022). Our work aims to avoid per-scene training, similar to PixelNeRF (Yu et al., 2021), IBRNet (Wang et al., 2021b), MVSNeRF (Chen et al., 2021a), and NeuRay (Liu et al., 2022) which train a cross-scene multi-view aggregator and reconstruct the radiance field with a one-shot forward pass.

**Transformer Meets Radiance Fields.**   Most similar to our work are NeRF methods that apply transformers for novel view synthesis and generalize across scenes. IBRNet (Wang et al., 2021b) processes sampled points on the ray using an MLP to predict color values and density features which are then input to a transformer to predict density. Recently, NeRFormer (Reizenstein et al., 2021) and Wang et al. (2022) use attention module to aggregate source views to construct feature volume with epipolar geometry constraints. However, a key difference with our work is that, all of them decode

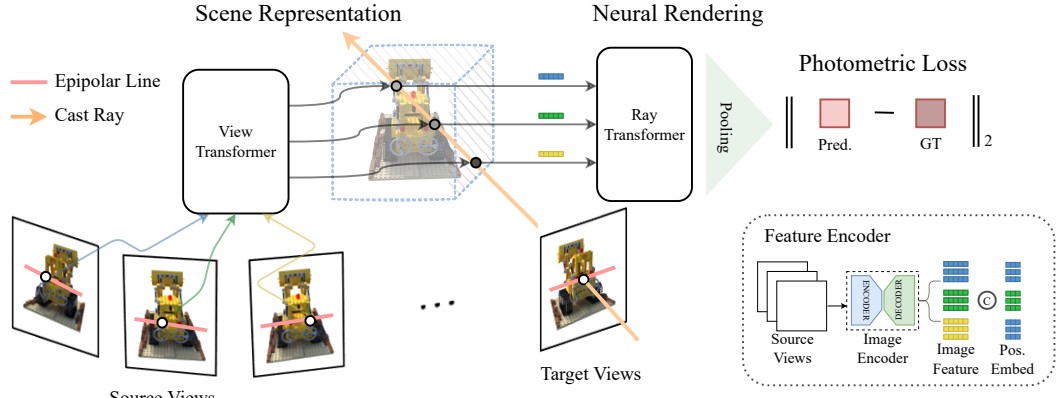

Figure 1: Overview of Generalizable NeRF Transformer (GNT): 1) Identify source views for a given target view, 2) Extract features for epipolar points using a trainable U-Net-like model, 3) For each ray in the target view, sample points and directly predict target pixel's color by aggregating view-wise features (View Transformer) and across points along a ray (Ray Transformer).

the latent feature representation to point-wise color and density, and rely on classic volume rendering to form the image, while our ray transformer learns to render the target pixel directly.

Other works, that use transformers but differ significantly in methodology or application, include (Lin et al., 2022) which generates novel views from just a single image via a vision transformer and SRT (Sajjadi et al., 2022b) which treats images and camera parameters in a latent space and trains a transformer that directly maps camera pose embedding to the corresponding image without any physical constraints. An alternative route formulates view synthesis as rendering a sparsely observed 4D light field, rather than following NeRF's 5D scene representation and volumetric rendering. The recently proposed NLF (Suhail et al., 2021) uses an attention-based framework to display light fields with view consistency, where the first transformer summarizes information on epipolar lines independently and then fuses epipolar features using a second transformer. This differs from GNT, where we aggregate across views, and are hence able to generalize across scenes which NLF fails to do. Lately, GPNR (Suhail et al., 2022), which was developed concurrently with our work, generalizes NLF (Suhail et al., 2021) by also enabling cross-view communication through the attention mechanism.

## 3 METHOD: MAKE ATTENTION ALL THAT NERF NEEDS

**Overview.**  Given a set of $N$ input views with known camera parameters $\{(\boldsymbol{I}_i \in \mathbb{R}^{H \times W \times 3}, \boldsymbol{P}_i \in \mathbb{R}^{3 \times 4})\}_{i=1}^{N}$, our goal is to synthesize novel views from arbitrary angles and also generalize to new scenes. Our method can be divided into two stages: (1) construct the 3D representation from source views on the fly in the feature space, (2) re-render the feature field at the specified angle to synthesize novel views. Unlike PixelNeRF, IBRNet, MVSNeRF and Neuray that borrow classic volume rendering for view synthesis after the first multi-view aggregation stage, we propose transformers to model both stages. Our pipeline is depicted in Fig. 1. First, the *view transformer* aggregates coordinate-aligned features from source views. To enforce multi-view geometry, we inject the inductive bias of epipolar constraints into the attention mechanism. After obtaining the feature representation of each point on the ray, the *ray transformer* composes point-wise features along the ray to form the ray color. This pipeline constitutes GNT and it is trained end-to-end.

### 3.1 EPIPOLAR GEOMETRY CONSTRAINED SCENE REPRESENTATION

NeRF represents 3D scene as a radiance field $\mathcal{F} : (\boldsymbol{x}, \boldsymbol{\theta}) \mapsto (\boldsymbol{c}, \sigma)$, where each spatial coordinate $\boldsymbol{x} \in \mathbb{R}^3$ together with the viewing direction $\boldsymbol{\theta} \in [-\pi, \pi]^2$ is mapped to a color $\boldsymbol{c} \in \mathbb{R}^3$ plus density $\sigma \in \mathbb{R}_+$ tuple. Vanilla NeRF parameterizes the radiance field using an MLP, and recovers the scene in a backward optimization fashion, inherently limiting NeRF from generalizing to new scenes. Generalizable NeRFs (Yu et al., 2021; Wang et al., 2021b; Chen et al., 2021a) construct the radiance

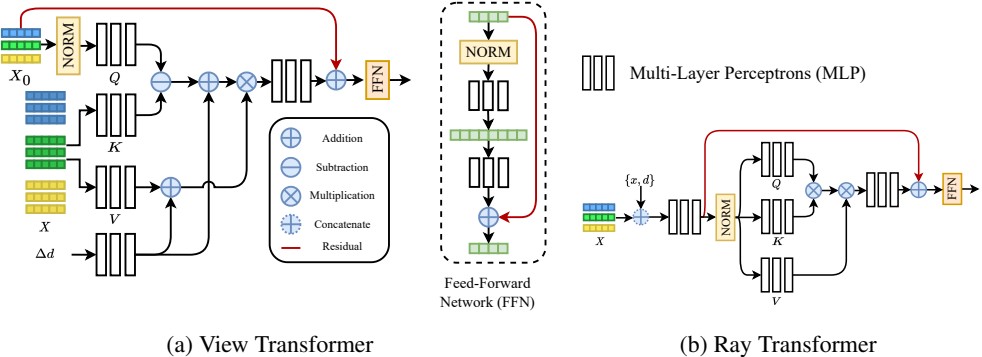

(a) View Transformer             (b) Ray Transformer

Figure 2: Detailed network architectures of view transformer and ray transformer in GNT, where $X$ represents the epipolar features, $X_0$ represents aggregated ray features, $\{x, d, \Delta d\}$ indicates point coordinates, viewing direction, and relative directions of source views with respect to the target view.

field in a feed-forward scheme, directly encoding multi-view images into 3D feature space, and decoding it to a color-density field.

In our work, we adopt the similar feed-forward fashion to convert multi-view images into 3D representation, but instead of using physical variables (e.g., color and density), we model a 3D scene as a coordinate-aligned feature field $\mathcal{F} : (\boldsymbol{x}, \boldsymbol{\theta}) \mapsto \boldsymbol{f} \in \mathbb{R}^d$, where $d$ is the dimension of the latent space. We formulate the feed-forward scene representation as follows:

$$\mathcal{F}(\boldsymbol{x}, \boldsymbol{\theta}) = \mathcal{V}(\boldsymbol{x}, \boldsymbol{\theta}; \{\boldsymbol{I}_1, \cdots, \boldsymbol{I}_N\}), \qquad (1)$$

where $\mathcal{V}(\cdot)$ is a function invariant to the permutation of input images to aggregate different views $\{\boldsymbol{I}_i, \cdots, \boldsymbol{I}_N\}$ into a coordinate-aligned feature field, and extracts features at a specific location. We use transformers as a set aggregation function (Lee et al., 2019). However, plugging in attention to globally attend to every pixel in the source images (Sajjadi et al., 2022b;a) is memory prohibitive and lacks multi-view geometric priors. Hence, we use epipolar geometry as an inductive bias that restricts each pixel to only attend to pixels that lie on the corresponding epipolar lines of the neighboring views. Specifically, we first encode each view to be a feature map $\boldsymbol{F}_i = \text{ImageEncoder}(\boldsymbol{I}_i) \in \mathbb{R}^{H \times W \times d}$. We expect the image encoder to extract not only shading information, but also material, semantics, and local/global complex light transport via its multi-scale architecture (Ronneberger et al., 2015). To obtain the feature representation at a position $\boldsymbol{x}$, we first project $\boldsymbol{x}$ to every source image, and interpolate the feature vector on the image plane. We then adopt a transformer (dubbed *view transformer*) to combine all the feature vectors. Formally, this process can be written as below:

$$\mathcal{F}(\boldsymbol{x}, \boldsymbol{\theta}) = \text{View-Transformer}(\boldsymbol{F}_1(\Pi_1(\boldsymbol{x}), \boldsymbol{\theta}), \cdots, \boldsymbol{F}_N(\Pi_N(\boldsymbol{x}), \boldsymbol{\theta})), \qquad (2)$$

where View-Transformer$(\cdot)$ is a transformer encoder (see Appendix A), $\Pi_i(\boldsymbol{x})$ projects $\boldsymbol{x} \in \mathbb{R}^3$ onto the $i$-th image plane by applying extrinsic matrix, and $\boldsymbol{F}_i(\boldsymbol{z}, \boldsymbol{\theta}) \in \mathbb{R}^d$ computes the feature vector at position $\boldsymbol{z} \in \mathbb{R}^2$ via bilinear interpolation on the feature grids. We use the transformer's positional encoding $\gamma(\cdot)$ to concatenate the extracted feature vector with point coordinate, viewing direction, and relative directions of source views with respect to the target view (similar to Wang et al. (2021b)). The detailed implementation of view transformer is depicted in Fig. 2. We defer our elaboration on its memory-efficient design to Appendix B. We argue that the view transformer can detect occlusion through the pixel values like a stereo-matching algorithm and selectively aggregate visible views (see details in Appendix E).

### 3.2 ATTENTION DRIVEN VOLUMETRIC RENDERING

Volume rendering (App. Eq. 7), which simulates outgoing radiance from a volumetric field has been regarded as a key knob of NeRF's success. NeRF renders the color of a pixel by integrating the color and density along the ray cast from that pixel. Existing works, including NeRF (Sec. 2), all use handcrafted and simplified versions of this integration. However, one can regard volume rendering as a weighted aggregation of all the point-wise output, in which the weights are globally dependent on the other points for occlusion modeling. This aggregation can be learned by a transformer such that point-wise colors can be mapped to token features, and attention scores correspond to transmittance (the blending weights). This is how we model the *ray transformer* which is illustrated in Fig. 2b

Table 1: Comparison of GNT against SOTA methods for single scene rendering. LLFF Dataset reports average scores on Orchids, Horns, Trex, Room, Leaves, Fern, Fortress.

| Models | PSNR↑ | SSIM↑ | LPIPS↓ | Avg↓ |
|---|---|---|---|---|
| LLFF | 24.88 | 0.911 | 0.114 | 0.051 |
| NeRF | 31.01 | 0.947 | 0.081 | 0.025 |
| MipNeRF | 33.09 | 0.961 | 0.043 | 0.016 |
| NLF | 33.85 | 0.981 | 0.024 | 0.011 |
| GNT | 33.71 | 0.975 | 0.025 | 0.011 |

| Models | PSNR↑ | SSIM↑ | LPIPS↓ | Avg↓ |
|---|---|---|---|---|
| LLFF | 23.93 | 0.798 | 0.212 | 0.896 |
| NeRF | 26.36 | 0.811 | 0.250 | 0.964 |
| NeX | 27.03 | 0.890 | 0.182 | 0.049 |
| NLF | 28.03 | 0.917 | 0.129 | 0.038 |
| GNT | 27.97 | 0.902 | 0.078 | 0.034 |

(a) NeRF Synthetic Dataset                          (b) Local Light Field Fusion (LLFF) Dataset

To render the color of a ray $r = (o, d)$, we can compute a feature representation $f_i = \mathcal{F}(x_i, \theta) \in \mathbb{R}^d$ for each point $x_i$ sampled on $r$. In addition to this, we also add position encoding of spatial location and view direction into $f_i$. We obtain the rendered color by feeding the sequence of $\{f_1, \cdots, f_M\}$ into the ray transformer, perform mean pooling over all the predicted tokens, and map the pooled feature vector to RGB via an MLP:

$$C(r) = \text{MLP} \circ \text{Mean} \circ \text{Ray-Transformer}(\mathcal{F}(o + t_1 d, \theta), \cdots, \mathcal{F}(o + t_M d, \theta)), \quad (3)$$

where $t_1, \cdots, t_M$ are uniformly sampled between near and far planes. Ray-Transformer is a standard transformer encoder, and its pseudocode implementation is provided in Appendix B. Rendering on feature space utilizes rich geometric, optical, and semantic information, which are intractable to be modeled explicitly. We argue that our ray transformer can automatically adjust the attention distribution to control the sharpness of the reconstructed surface, and bake desirable lighting effects from the illumination and material features. Moreover, by exerting the expressiveness of the image encoder, the ray transformer can also overcome the limitation of ray casting and epipolar geometry to simulate complex light transport (e.g., refraction, reflection, etc.). Interestingly, despite all in latent space, we can also infer some explicit physical properties (such as depth) from ray transformer. See Appendix E for depth cueing. We also involve discussion on the extension to auto-regressive rendering and attention-based coarse-to-fine sampling in Appendix C.

## 4 EXPERIMENTS

We conduct experiments to compare GNT against state-of-the-art methods for novel view synthesis. Our experiment settings include both *per-scene* optimization and *cross-scene* generalization.

### 4.1 IMPLEMENTATION DETAILS

**Source and Target View Sampling.**    Following IBRNet, we construct pairs of source and target views for training by first selecting a target view, and then identifying a pool of $k \times N$ nearby views, from which $N$ views are randomly sampled as source views. This sampling strategy simulates various view densities during training and therefore helps the network generalize better. During training, the values for $k$ and $N$ are uniformly sampled at random from $(1, 3)$ and $(8, 12)$ respectively.

**Training / Inference Details.**    We train both the feature extraction network and GNT end-to-end on datasets of multi-view posed images using the Adam optimizer to minimize the mean-squared error between predicted and ground truth RGB pixel values. The base learning rates for the feature extraction network and GNT are $10^{-3}$ and $5 \times 10^{-4}$ respectively, which decay exponentially over training steps. For all our experiments, we train for 250,000 steps with 4096 rays sampled in each iteration. Unlike most NeRF methods, we do not use separate coarse, fine networks and therefore to bring GNT to a comparable experimental setup, we sample 192 coarse points per ray across all experiments (unless otherwise specified).

**Metrics.**    We use three widely adopted metrics: Peak Signal-to-Noise Ratio (PSNR), Structural Similarity Index Measure (SSIM) (Wang et al., 2004), and the Learned Perceptual Image Patch Similarity (LPIPS) (Zhang et al., 2018). We report the averages of each metric over different views in each scene (for single-scene experiments) and across multiple scenes in each dataset (for generalization experiments). We additionally report the geometric mean of $10^{-PSNR/10}$, $\sqrt{1 - SSIM}$ and LPIPS, which provides a summary of the three metrics for easier comparison (Barron et al., 2021).

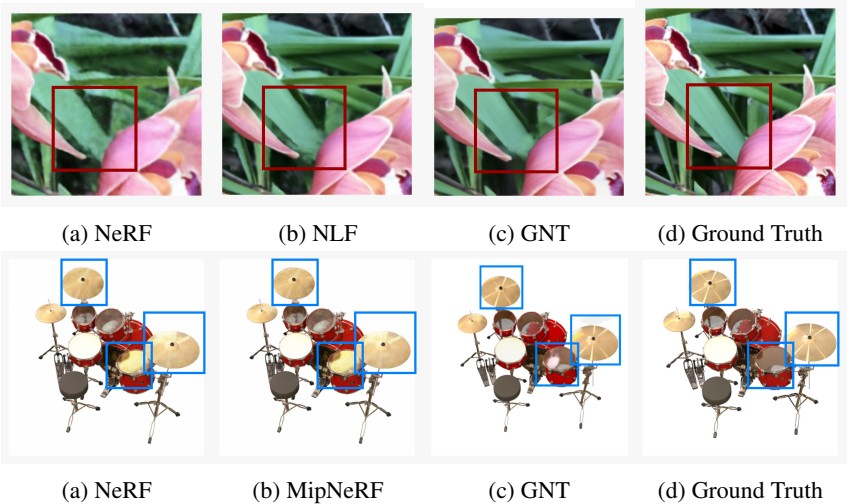

| (a) NeRF | (b) NLF | (c) GNT | (d) Ground Truth |

| (a) NeRF | (b) MipNeRF | (c) GNT | (d) Ground Truth |

Figure 3: Qualitative results for single-scene rendering. In the Orchids scene from LLFF (first row), GNT recovers the shape of the leaves more accurately. In the Drums scene from Blender (second row), GNT's learnable renderer is able to model physical phenomena like specular reflections.

## 4.2    SINGLE SCENE RESULTS

**Datasets.**    To evaluate the single scene view generation capacity of GNT, we perform experiments on datasets containing synthetic rendering of objects and real images of complex scenes. In these experiments, we use the same resolution and train/test splits as NeRF (Mildenhall et al., 2020). **Local Light Field Fusion (LLFF) dataset**: Introduced by Mildenhall et al. (2019), it consists of 8 forward facing captures of real-world scenes using a smartphone. We report average scores across {Orchids, Horns, Trex, Room, Leaves, Fern, Fortress} and the metrics are summarized in Tab. 1b. **NeRF Synthetic Dataset**: The synthetic dataset introduced by (Mildenhall et al., 2020) consists of 8, 360°scenes of objects with complicated geometry and realistic material. Each scene consists of images rendered from viewpoints randomly sampled on a hemisphere around the object. Similar to the experiments on the LLFF dataset, we report the average metrics across all eight scenes in Tab. 1a.

**Discussion.**    We compare our GNT with LLFF , NeRF , MipNeRF , NeX , and NLF. Compared to other methods, we utilize a smaller batch size (specifically GNT samples 4096 rays per batch while NLF samples as much as 16384 rays per batch) and only sample coarse points fed into the network in one single forward pass, unlike most methods that use a two-stage coarse-fine sampling strategy. These hyperparameters have a strong correlation with the rendered image quality, leaving our method at a disadvantage. Despite these differences, GNT still manages to outperform most methods and performs on par when compared to SOTA NLF method on both LLFF and Synthetic datasets. We provide a scene-wise breakdown of results on both these datasets in Appendix D (Tab. 6, 7). In complex scenes like Drums, Ship, and Leaves, GNT manages to outperform other methods more substantially by 2.49 dB, 0.82 dB and 0.10 dB respectively. This indicates the effectiveness of our attention-driven volumetric rendering to model complex conditions. Interestingly, even in the worse performing scenes by PSNR (e.g. T-Rex), GNT achieves best perceptual metric scores across all scenes in the LLFF dataset (i.e LPIPS ~27% ↓). This could be because PSNR fails to measure structural distortions, blurring, has high sensitivity towards brightness, and hence does not effectively measure visual quality. Similar inferences are discussed in Lin et al. (2022) regarding discrepancies in PSNR scores and their correlation to rendered image quality. Fig. 3 provides qualitative comparisons on the Orchids, Drums scene respectively and we can clearly see that GNT recovers the edge details of objects (in the case of Orchids) and models complex lighting effect like specular reflection (in the case of Drums) more accurately.

## 4.3    GENERALIZATION TO UNSEEN SCENES

**Datasets.**    GNT leverages multi-view features complying with epipolar geometry, enabling generalization to unseen scenes. We follow the experimental protocol in IBRNetto evaluate the cross-scene generalization of GNT and use the following datasets for training and evaluation, respectively.

Table 2: Comparison of GNT against SOTA methods for cross-scene generalization.

| Models | Local Light Field Fusion (LLFF) | | | | NeRF Synthetic | | | |
|---|---|---|---|---|---|---|---|---|
| | PSNR↑ | SSIM↑ | LPIPS↓ | Avg↓ | PSNR↑ | SSIM↑ | LPIPS↓ | Avg↓ |
| PixelNeRF | 18.66 | 0.588 | 0.463 | 0.159 | 22.65 | 0.808 | 0.202 | 0.078 |
| MVSNeRF | 21.18 | 0.691 | 0.301 | 0.108 | 25.15 | 0.853 | 0.159 | 0.057 |
| IBRNet | 25.17 | 0.813 | 0.200 | 0.064 | 26.73 | 0.908 | 0.101 | 0.040 |
| NeuRay | 25.35 | 0.818 | 0.198 | 0.062 | 28.29 | 0.927 | 0.080 | 0.032 |
| GPNR | 25.72 | 0.880 | 0.175 | 0.055 | 26.48 | 0.944 | 0.091 | 0.036 |
| GNT | 25.86 | 0.867 | 0.116 | 0.047 | 27.29 | 0.937 | 0.056 | 0.029 |

(a) Blender and LLFF Datasets

| Setting | Models | Shiny-6 Dataset | | | |
|---|---|---|---|---|---|
| | | PSNR↑ | SSIM↑ | LPIPS↓ | Avg↓ |
| Per-Scene Training | NeRF | 25.60 | 0.851 | 0.259 | 0.065 |
| | NeX | 26.45 | 0.890 | 0.165 | 0.049 |
| | IBRNet | 26.50 | 0.863 | 0.122 | 0.047 |
| | NLF | 27.34 | 0.907 | 0.045 | 0.029 |
| Generalization | IBRNet | 23.60 | 0.785 | 0.180 | 0.071 |
| | GPNR | 24.12 | 0.860 | 0.170 | 0.063 |
| | GNT | 27.10 | 0.912 | 0.083 | 0.036 |

(b) Shiny Dataset

(a) **Training Datasets** consists of both real, and synthetic data. For synthetic data, we use object renderings of 1023 models from Google Scanned Object (Downs et al., 2022). For real data, we make use of RealEstate10K (Zhou et al., 2018b), 100 scenes from the Spaces dataset (Flynn et al., 2019), and 102 real scenes from handheld cellphone captures (Mildenhall et al., 2019; Wang et al., 2021b).

(b) **Evaluation Datasets** include the previously discussed Synthetic (Mildenhall et al., 2020), LLFF datasets (Mildenhall et al., 2019) and Shiny-9 dataset (Wizadwongsa et al., 2021) with complex optics. Please note that the LLFF scenes present in the validation set are not included in the handheld cellphone captures in the training set.

**Discussion.**    We compare our method with PixelNeRF (Yu et al., 2021), MVSNeRF (Chen et al., 2021a), IBRNet (Wang et al., 2021b), and NeuRay (Liu et al., 2022). As seen from Tab. 2a, our method outperforms SOTA by ~17% ↓, ~9% ↓ average scores in the LLFF, Synthetic datasets respectively. This indicates the effectiveness of our proposed view transformer to extract generalizable scene representations. Similar to the single scene experiments, we observe significantly better perceptual metric scores 3% ↑ SSIM, 27% ↓ LPIPS in both datasets. We show qualitative results in Fig. 5 where GNT renders novel views with clearly better visual quality when compared to other methods. Specifically, as seen from the second row in Fig. 5, GNT is able to handle regions that are sparsely visible in the source views and generates images of comparable visual quality as NeuRay even with no explicit supervision for occlusion. We also provide an additional comparison against SRT (Sajjadi et al., 2022b) in Appendix D (Tab. 5), where our GNT significantly generalizes better.

**GNT can learn to adapt to refraction and reflection in scenes.**    Encouraged by the promise shown by GNT in modeling reflections in the Drums scene via pre-scene training, we further directly evaluate pre-trained GNT on the Shiny dataset (Wizadwongsa et al., 2021), which contains several challenging view-dependent effects, such as the rainbow reflections on a CD, and the refraction through liquid bottles. Technically, the full formulation of volume rendering (radiative transfer equation, as used in modern volume path tracers) is capable of handling all these effects. However, standard NeRFs use a simplified formulation which does not simulate all physical effects, and hence easily fail to capture these effects.

Tab. 2b presents the numerical results of GNT when generalizing to Shiny dataset. Notably, our GNT outperforms state-of-the-art GPNR (Suhail et al., 2022) by 3dB in PSNR and ~40% in average metric. Compared with per-scene optimized NeRFs, GNT outperforms many of them and even approaches to the best performer NLF (Suhail et al., 2021) without any extra training. This further supports our argument that cross-scene training can help learn a better renderer. Fig. 4 exhibits rendering results on the two example scenes of Lab and CD. Compared to the baseline (NeX), GNT is able to reconstruct complex refractions through test tube, and the interference patterns on the disk with higher quality, indicating the strong flexibility and adaptivity of our learnable renderer. That "serendipity" is intriguing to us, since the presence of refraction and scattering means that any technique that only uses samples along a single ray will not be able to properly simulate the full light transport. We conjecture GNT's success in modeling those challenging physical scenes to be the full transformer

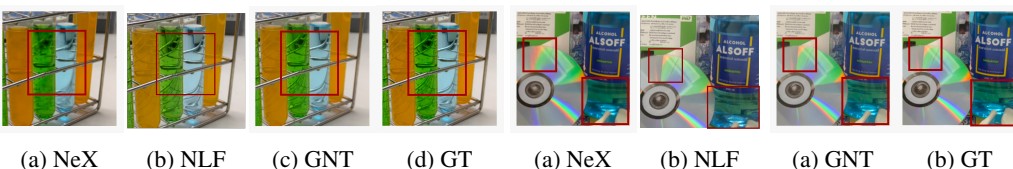

| (a) NeX | (b) NLF | (c) GNT | (d) GT | (a) NeX | (b) NLF | (a) GNT | (b) GT |

Figure 4: Qualitative results of GNT for generalizable rendering on the the complex Shiny dataset, that contains more refractions and reflection. A pre-trained GNT can naturally adapt to complex refractions through test tube, and the interference patterns on the disk with higher quality.

architectures for not only rays but also views: the view encoder aggregates multi-ray information and can in turn tune the ray encoding itself.

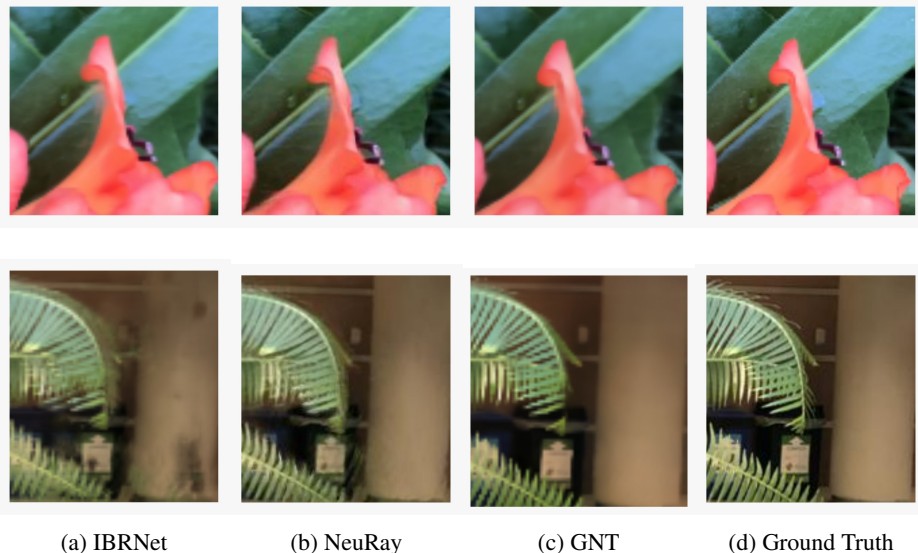

| (a) IBRNet | (b) NeuRay | (c) GNT | (d) Ground Truth |

Figure 5: Qualitative results for the cross-scene rendering. On the unseen Flowers (first row) and Fern (second row) scenes, GNT recovers the edges of petals and pillars more accurately than IBRNet and visually comparable to NeuRay.

## 4.4 ABLATION STUDIES

We conduct the following ablation studies on the Drums scene to validate our architectural designs.

**One-Stage Transformer**: We convert the point-wise epipolar features into one single sequence and pass it through a "one-stage transformer" network with standard dot-product self-attention layers, without considering our two-stage pipeline: view and ray aggregation.

**Epipolar Agg. → View Agg.**: Moving to a two-stage transformer, we train a network that first aggregates features from the points along the epipolar lines followed by feature aggregation across epipolar lines on different reference views (in contrast to GNT's first view aggregation then ray aggregation). This two-stage aggregation resembles the strategies adopted in NLF (Suhail et al., 2021).

Table 3: Ablation study of several components in GNT on the Drums scene from the Blender dataset. The indent indicates the studied setting is added upon the upper-level ones.

| Model | PSNR↑ | SSIM↑ | LPIPS↓ | Avg↓ |
|---|---|---|---|---|
| One-Stage Transformer | 21.80 | 0.862 | 0.152 | 0.072 |
| Two-Stage Transformer | | | | |
|   Epipolar Agg. → View Agg. | 21.57 | 0.863 | 0.153 | 0.073 |
|   View Agg. → Ray Agg. | | | | |
|     Dot Product-Attention View Transformer | 26.98 | **0.953** | 0.089 | 0.034 |
|     Subtraction-Attention View Transformer | | | | |
|       w/ Volumetric Rendering | 24.24 | 0.92 | 0.076 | 0.043 |
|       Learned Volumetric Rendering (Ours) | **29.41** | 0.931 | **0.085** | **0.029** |

**Dot Product-Attention View Transformer**: Next we train a network that uses the standard dot-product attention in the view transformer blocks, in contrast to our proposed memory-efficient subtraction-based attention (see Appendix B).

**w/ Volumetric Rendering**: Last but not least, we train a network to predict per-point RGB and density values from the point feature output by the view aggregator, and compose them using the volumetric rendering equation, instead of our learned attention-driven volumetric renderer.

We report the performance of the above investigation in Tab. 3. We verify that our design of two-stage transformer is superior to one-stage aggregation or an alternative two-stage pipeline (Suhail et al., 2021) since our renderer strictly complies with the multi-view geometry. Compared with dot-product attention, subtraction-based attention achieves slightly higher overall scores. This also indicates the performance of GNT does not heavily rely on the choice of attention operation. What matters is bringing in the attention mechanism for cross-point interaction. For practical usage, we also consider the memory efficiency in our view transformer. Our ray transformer outperforms the classic volumetric rendering, implying the advantage of adopting a data-driven renderer.

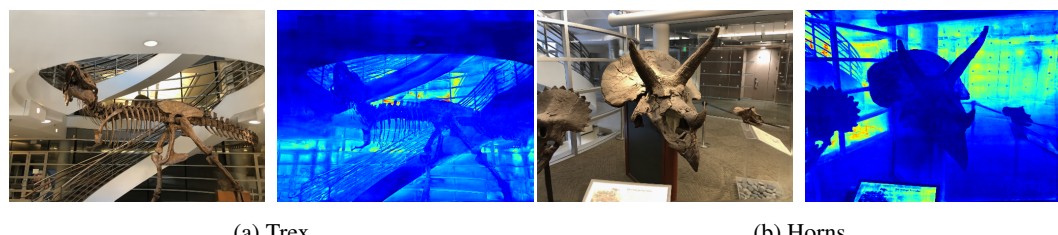

(a) Trex                                          (b) Horns

Figure 7: Visualization of ray attention where each color indicates the distance of each pixel relative to the viewing direction. GNT's ray transformer computes point-wise aggregation weights from which the depth can be inferred. Red indicates far while blue indicates near.

## 4.5 INTERPRETATION ON ATTENTION MAPS

The use of transformers as a core element in GNT enables interpretation by analyzing the attention weights. As discussed earlier, view transformer finds correspondence between the queried points, and neighboring views which enables it to pay attention to more "visible" views or be occlusion-aware. Similarly, the ray transformer captures point-to-point interactions which enable it to model the relative importance of each point or be depth-aware. We validate our hypothesis by visualization.

**View Attention.** To visualize the view-wise attention maps learned by our model, we use the attention matrix from Eq. 9 and collapse the channel dimension by performing mean pooling. We then identify the view number which is assigned maximum attention with respect to each point and then compute the most repeating view number across points along a ray (by computing mode). These "view importance" values denote the view which has maximum correspondence

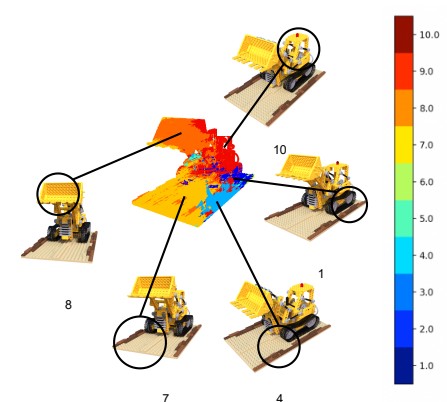

Figure 6: Visualization of view attention where each color indicates the view number that has maximum correspondence with a target pixel. GNT's view transformer learns to map each object region in the target view to its corresponding regions in the source views which are least occluded.

with the target pixel's color. Fig. 6 visualizes the source view correspondence with every pixel in the target view. Given a region in the target view, GNT attempts to pay maximum attention to a source view that is least occluded in the same region. For example: In Fig. 6, the truck's bucket is most visible from view number 8, hence the regions corresponding to the same are orange colored, while regions towards the front of the lego are most visible from view number 7 (yellow).

**Ray Attention.** To visualize the attention maps across points in a ray, we use the attention matrix from Eq. 4 and collapse the head dimension by performing mean pooling. From the derived matrix, we select a point and extract its relative importance with every other point. We then compute a depth map from these learned point-wise correspondence values by multiplying it with the marching distance of each point and sum-pooling along a ray. Fig. 7 plots the depth maps computed from the learned attention values in the ray transformer block. We can clearly see that the obtained depth maps have a physical meaning i.e pixels closer to the view directions are blue while the ones farther away are red. Therefore, with no explicit supervision, GNT learns to physically ground its attention maps.

## 5 CONCLUSION

We present Generalizable NeRF Transformer (GNT), a pure transformer-based architecture that efficiently reconstructs NeRFs on the fly. The view transformer of GNT leverages epipolar geometry as an inductive bias for scene representation. The ray transformer renders novel views by ray marching and decoding the sequences of sampled point features using the attention mechanism. Extensive experiments demonstrate that GNT improves both single-scene and cross-scene training results, and demonstrates "out of the box" promise for refraction and reflection scenes. We also show by visualization that depth and occlusion can be inferred from attention maps. This implies that pure attention can be a "universal modeling tool" for the physically-grounded rendering process. Future directions include relaxing the epipolar constraints to simulate more complicated light transport.

ACKNOWLEDGMENTS

We thank Pratul Srinivasan for his comments on a draft of this work.

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

## A   PRELIMINARIES

**Self-Attention and Transformer.**   Multi-Head Self-Attention (MHA) is the key ingredient of transformers (Vaswani et al., 2017). Data is first tokenized into sequences and a pairwise score is computed to weight the relation of each token with all the others in a given input context. Formally, let $\boldsymbol{X} \in \mathbb{R}^{N \times d}$ represent some sequential data with $N$ tokens of $d$-dimension. A self-attention layer transforms the feature matrix as below:

$$\text{Attn}(\boldsymbol{X}) = \text{softmax}(\boldsymbol{A}) f_V(\boldsymbol{X}), \text{ where } \boldsymbol{A}_{i,j} = \alpha(\boldsymbol{X}_i, \boldsymbol{X}_j), \forall i, j \in [N] \tag{4}$$

where $\boldsymbol{A} \in \mathbb{R}^{N \times N}$ is called the attention matrix, $\text{softmax}(\cdot)$ operation normalizes the attention matrix row-wise, and $\alpha(\cdot)$ represents a pair-wise relation function, most commonly the dot-product $\alpha(\boldsymbol{X}_i, \boldsymbol{X}_j) = f_Q(\boldsymbol{X}_i)^\top f_K(\boldsymbol{X}_j)/\gamma$, where $f_Q(\cdot), f_K(\cdot), f_V(\cdot)$ are called query, key, and value mapping functions. In a standard transformer, they are chosen as fully-connected layers. This self-attention is akin to an aggregation operation. Multi-Head Self-Attention (MHA) sets a group of self-attention blocks, and adopts a linear layer to project them onto the output space:

$$\text{MHA}(\boldsymbol{X}) = [\text{Attn}_1(\boldsymbol{X}) \quad \text{Attn}_2(\boldsymbol{X}) \quad \cdots \quad \text{Attn}_H(\boldsymbol{X})] \boldsymbol{W}_O \tag{5}$$

Following an MHA block, one standard layer of transformer also adopts a Feed-Forward Network (FFN) to do point-wise feature transformation, as well as skip connection and layer normalization to stablize training. The whole transformer block can be formulated as below:

$$\tilde{\boldsymbol{X}} = \text{MHA}(\text{LayerNorm}(\boldsymbol{X})) + \boldsymbol{X}, \quad \boldsymbol{Y} = \text{FFN}(\text{LayerNorm}(\tilde{\boldsymbol{X}})) + \tilde{\boldsymbol{X}} \tag{6}$$

**Neural Radiance Field.**   NeRFs (Mildenhall et al., 2020) converts multi-view images into a radiance field and interpolates novel views by re-rendering the radiance field from a new angle. Technically, NeRF models the underlying 3D scene as a continuous radiance field $\mathcal{F} : (\boldsymbol{x}, \boldsymbol{\theta}) \mapsto (\boldsymbol{c}, \sigma)$ parameterized by a Multi-Layer Perceptron (MLP) $\boldsymbol{\Theta}$, which maps a spatial coordinate $\boldsymbol{x} \in \mathbb{R}^3$ together with the viewing direction $\boldsymbol{\theta} \in [-\pi, \pi]^2$ to a color $\boldsymbol{c} \in \mathbb{R}^3$ plus density $\sigma \in \mathbb{R}_+$ tuple. To form an image, NeRF performs the ray-based rendering, where it casts a ray $\boldsymbol{r} = (\boldsymbol{o}, \boldsymbol{d})$ from the optical center $\boldsymbol{o} \in \mathbb{R}^3$ through each pixel (towards direction $\boldsymbol{d} \in \mathbb{R}^3$), and then leverages volume rendering (Max, 1995) to compose the color and density along the ray between the near-far planes:

$$\boldsymbol{C}(\boldsymbol{r}|\boldsymbol{\Theta}) = \int_{t_n}^{t_f} T(t)\sigma(\boldsymbol{r}(t))\boldsymbol{c}(\boldsymbol{r}(t), \boldsymbol{d})dt, \text{ where } T(t) = \exp\left(-\int_{t_n}^t \sigma(s)ds\right), \tag{7}$$

where $\boldsymbol{r}(t) = \boldsymbol{o} + t\boldsymbol{d}$, $t_n$ and $t_f$ are the near and far planes respectively. In practice, the Eq. 7 is numerically estimated using quadrature rules (Mildenhall et al., 2020). Given images captured from surrounding views with known camera parameters, NeRF fits the radiance field by maximizing the likelihood of simulated results. Suppose we collect all pairs of rays and pixel colors as the training set $\mathcal{D} = \{(\boldsymbol{r}_i, \widehat{\boldsymbol{C}}_i)\}_{i=1}^N$, where $N$ is the total number of rays sampled, and $\widehat{\boldsymbol{C}}_i$ denotes the ground-truth color of the $i$-th ray, then we train the implicit representation $\boldsymbol{\Theta}$ via the following loss function:

$$\mathcal{L}(\boldsymbol{\Theta}|\mathcal{R}) = \mathbb{E}_{(\boldsymbol{r}, \widehat{\boldsymbol{C}}) \in \mathcal{D}} \|\boldsymbol{C}(\boldsymbol{r}|\boldsymbol{\Theta}) - \widehat{\boldsymbol{C}}(\boldsymbol{r})\|_2^2, \tag{8}$$

## B   IMPLEMENTATION DETAILS

**Memory-Efficient Cross-View Attention.**   Computing attention between every pair of inputs has $O(N^2)$ memory complexity, which is computational prohibitive when sampling thousands of points at the same time. Nevertheless, we note that view transformer only needs to read out one token as the fused results of all the views. Therefore, we propose to only place one read-out token $\boldsymbol{X}_0 \in \mathbb{R}^d$ in the query sequence, and let it iteratively summarize features from other data points. This reduces the complexity for each layer up to $O(N)$. We initialize the read-out token as the element-wise max-pooling of all the inputs: $\boldsymbol{X}_0 = \max(\boldsymbol{F}_1(\Pi_1(\boldsymbol{x}), \boldsymbol{\theta}), \cdots, \boldsymbol{F}_N(\Pi_N(\boldsymbol{x}), \boldsymbol{\theta}))$. Rather than adopting a standard dot-product attention, we choose subtraction operation as the relation function. Subtraction attention has been shown more effective for positional and geometric relationship reasoning (Zhao et al., 2021; Fan et al., 2022). Compared with dot-product that collapses the feature dimension into a scalar, subtraction attention computes different attention scores for every channel of the value matrix, which increases diversity in feature interactions. Moreover, we augment the attention map and value

matrix with $\{\boldsymbol{\Delta d}\}_{i=1}^{N}$ to provide relative spatial context. Technically, we utilize a linear layer $\boldsymbol{W}_P$ to lift $\boldsymbol{\Delta d}_i$ to the hidden dimension. We illustrate view transformer in Fig. 2a. To be more specific, the modified attention adopted in our view transformer can be formulated as:

$$\text{View-Attn}(\boldsymbol{X}) = \text{diag}(\text{softmax}(\boldsymbol{A} + \boldsymbol{\Delta}^{\top})f_V(\boldsymbol{X} + \boldsymbol{\Delta})), \text{ where } \boldsymbol{A}_j = f_Q(\boldsymbol{X}_0) - f_K(\boldsymbol{X}_j), \tag{9}$$

where $\boldsymbol{A}_j \in \mathbb{R}^d$ denotes the $j$-th column of $\boldsymbol{A}$, $\boldsymbol{\Delta} = [\boldsymbol{\Delta d}_1 \quad \cdots \quad \boldsymbol{\Delta d}_N]^{\top} \boldsymbol{W}_P \in \mathbb{R}^{N \times d}$, $f_Q$, $f_K$, and $f_V$ are parameterized by an MLP. We note that by applying $\text{diag}(\cdot)$, we read out the updated query token $\boldsymbol{X}_0$. See Alg. 1 for the implementation in practice.

**Network Architecture.** To extract features from the source views, we use a U-Net-like architecture with a ResNet34 encoder, followed by two up-sampling layers as decoder.Each view transformer block contains a single-headed cross-attention layer while the ray transformer block contains a multi-headed self-attention layer with four heads. The outputs from these attention layers are passed onto corresponding feedforward blocks with a Rectified Linear Unit (RELU) activation and a hidden dimension of 256. A residual connection is applied between the pre-normalized inputs (LayerNorm) and outputs at each layer. For all our single-scene experiments, we alternatively stack 4 view and ray transformer blocks while our larger generalization experiments use 8 blocks each. All transformer blocks (view and ray) are of dimension 64. Following Vaswani et al. (2017); Mildenhall et al. (2020); Zhong et al. (2021), we convert the low-dimensional coordinates to a high-dimensional representation using Fourier components, where the number of frequencies is selected as 10 for all our experiments. The derived view and position embeddings are each of dimension 63.

---

**Algorithm 1** Cross View Attention: PyTorch-like Pseudocode

---

$\boldsymbol{X}_0 \rightarrow$ coordinate aligned features$(N_{\text{rays}}, N_{\text{pts}}, D)$
$\boldsymbol{X}_j \rightarrow$ epipolar view features$(N_{\text{rays}}, N_{\text{pts}}, N_{\text{views}}, D)$
$\boldsymbol{\Delta d} \rightarrow$ relative directions of source views wrt target views$(N_{\text{rays}}, N_{\text{pts}}, N_{\text{views}}, 3)$
$f_Q, f_K, f_V, f_P, f_A, f_O \rightarrow$ functions that parameterize MLP layers

$\boldsymbol{Q} = f_Q(\boldsymbol{X}_0)$
$\boldsymbol{K} = f_K(\boldsymbol{X}_j)$
$\boldsymbol{V} = f_V(\boldsymbol{X}_j)$

$\boldsymbol{P} = f_P(\boldsymbol{\Delta d})$
$\boldsymbol{A} = \boldsymbol{K} - \boldsymbol{Q}[:, :, \text{None}, :] + \boldsymbol{P}$
$A = \text{softmax}(\boldsymbol{A}, \text{dim} = -2)$

$\boldsymbol{O} = ((\boldsymbol{V} + \boldsymbol{P}) \cdot \boldsymbol{A}).\text{sum}(\text{dim} = 2)$
$\boldsymbol{O} = f_O(\boldsymbol{O})$

---

**Pseudocode.** We provide a simple and efficient pytorch pseudo-code to implement the attention operations in the view, ray transformer blocks in Alg. 1, 2 respectively. We do not indicate the feedforward and layer normalization operations for simplicity. As seen in Alg. 3, we reuse the epipolar view features $X_j$ to derive keys, and values across view transformer blocks. Therefore, one could further improve efficiency by computing them only once while also sharing the network weights across view transformer blocks or simply put $f_{\text{view }i}(.)$ represents the same function across different values of $i$. This can be considered analogous to an unfolded recurrent neural network that updates itself iteratively but using the same weights.

## C    TENTATIVE EXTENSIONS

### C.1    AUTO-REGRESSIVE DECODING

The final rendered color is obtained by mean-pooling the outputs from the ray transformer block, and mapping the pooled feature vector to RGB via an MLP layer. It can be understood that the target pixel's color is strongly dependent on the closest point from the ray origin and weakly related to the

---

**Algorithm 2** Ray Attention: PyTorch-like Pseudocode

---

$\boldsymbol{X}_0 \to$ coordinate aligned features$(N_{\text{rays}}, N_{\text{pts}}, D)$
$\boldsymbol{x} \to$ point coordinates (after position encoding)$(N_{\text{rays}}, N_{\text{pts}}, D)$
$\boldsymbol{d} \to$ target view direction (after position encoding)$(N_{\text{rays}}, N_{\text{pts}}, D)$
$f_Q, f_K, f_V, f_P, f_A, f_O \to$ functions that parameterize MLP layers

$\boldsymbol{X}_0 = f_P(\text{concat}(X_o, d, x))$
$\boldsymbol{Q} = f_Q(\boldsymbol{X}_0)$
$\boldsymbol{K} = f_K(\boldsymbol{X}_0)$
$\boldsymbol{V} = f_V(\boldsymbol{X}_0)$

$\boldsymbol{A} = \text{matmul}(\boldsymbol{Q}, \boldsymbol{K}^T)/\sqrt{D}$
$\boldsymbol{A} = \text{softmax}(\boldsymbol{A}, \dim = -1)$

$\boldsymbol{O} = \text{matmul}(\boldsymbol{V}, \boldsymbol{A})$
$\boldsymbol{O} = f_O(\boldsymbol{O})$

---

**Algorithm 3** GNT: PyTorch-like Pseudocode

---

$\boldsymbol{X}_j \to$ epipolar view features$(N_{\text{rays}}, N_{\text{pts}}, N_{\text{views}}, D)$
$\boldsymbol{x} \to$ point coordinates (after position encoding)$(N_{\text{rays}}, N_{\text{pts}}, D)$
$\boldsymbol{d} \to$ target view direction (after position encoding)$(N_{\text{rays}}, N_{\text{pts}}, D)$
$\boldsymbol{\Delta d} \to$ relative directions of source views wrt target views$(N_{\text{rays}}, N_{\text{pts}}, N_{\text{views}}, 3)$
$f_{\text{view}}^{(l)}, f_{\text{ray}}^{(l)} \to$ functions that parameterize view transforms, ray transformers at layer $l$ respectively
$f_{\text{rgb}} \to$ functions that parameterize MLP layers

$l = 0$
$\boldsymbol{X}_0 = \max_{i=1}^{N_{\text{views}}}(\boldsymbol{X}_j)$
**while** $l < N_{\text{layers}}$ **do**
    $\boldsymbol{X}_0 = f_{\text{view}}^{(l)}(\boldsymbol{X}_0, \boldsymbol{X}_j, \boldsymbol{\Delta d})$
    $\boldsymbol{X}_0 = f_{\text{ray}}^{(l)}(\boldsymbol{X}_o, \boldsymbol{d}, \boldsymbol{x})$
    $l = l + 1$
**end while**
$\boldsymbol{O} = \text{Norm}(\boldsymbol{X}_0)$
$\text{RGB} = f_{\text{rgb}}(\text{mean}_{i=1}^{N_{\text{points}}}(\boldsymbol{X}_0))$

---

farthest. Revisiting Eq. 7, volumetric rendering attempts to compose point-wise color depending on the other points in a far to near fashion. Motivated by this, we propose an auto-regressive decoder to better simulate the rendering process. Transformers have shown great success in auto-regressive decoding, more specifically in NLP (Vaswani et al., 2017). We borrow a similar strategy and replace the simpler MLP-based color prediction with a series of transformer blocks - with self, cross attention layers.

In the first pass, the decoder is queried with positional encoding of the farthest point ($\gamma(\boldsymbol{x}_N)$) to generate an output feature representation of the same. In the next step, the output token is concatenated with the second farthest point ($\gamma(\boldsymbol{x}_{N-1})$) to query the decoder. This process repeats until all the points in the ray are queried in a far-to-near fashion. In the final pass, the encoded view direction ($\gamma(\boldsymbol{d})$) is concatenated with the per-point output features in the previous passes to query the decoder and the output token corresponding to the view direction is extracted. The extracted token is mapped to RGB via an MLP layer. This entire process is summarized in Fig. 8. The auto-regressive procedure closely resembles the volumetric rendering equation which iteratively blends and overrides the previous color when marching along a ray from far to near.

Transformer-based decoders used in language iteratively predict output tokens only during inference, i.e. they are trained in a non-autoregressive fashion due to the availability of ground truth output tokens in each step. Transferring the same to neural rendering is not possible, as we do not have access to the groundtruth color for each point sampled along the ray. Hence, we require a loop to

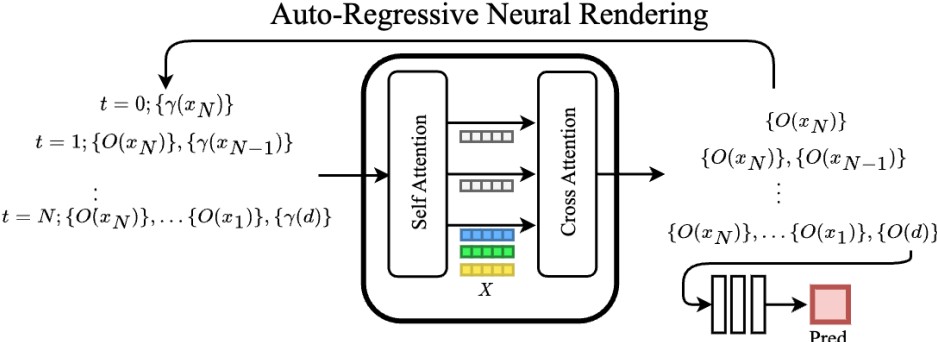

Figure 8: Architecture of auto-regressive ray decoder with sampling strategy in a far to near fashion.

Table 4: Comparison of autoregressive GNT against SOTA methods for single scene rendering on the LLFF dataset.

| Models | Orchids | | | | Horns | | | |
|---|---|---|---|---|---|---|---|---|
| | PSNR↑ | SSIM↑ | LPIPS↓ | Avg↓ | PSNR↑ | SSIM↑ | LPIPS↓ | Avg↓ |
| LLFF | 18.52 | 0.588 | 0.313 | 0.141 | 23.22 | 0.840 | 0.193 | 0.064 |
| NeRF | 20.36 | 0.641 | 0.321 | 0.121 | 27.45 | 0.828 | 0.268 | 0.058 |
| NeX | 20.42 | 0.765 | 0.242 | 0.102 | 28.46 | 0.934 | 0.173 | 0.040 |
| NLF | 21.05 | 0.807 | 0.173 | 0.084 | 29.78 | 0.957 | 0.121 | 0.030 |
| GNT | 20.67 | 0.752 | 0.153 | 0.087 | 29.62 | 0.935 | 0.076 | 0.028 |
| GNT + AutoReg | 21.05 | 0.736 | 0.181 | 0.090 | 28.20 | 0.908 | 0.114 | 0.037 |
| GNT + Fine | 20.69 | 0.752 | 0.153 | 0.087 | 29.59 | 0.934 | 0.076 | 0.028 |

auto-regressively decode features even during training. This reduces the computational efficiency of the proposed strategy, especially as the number of points sampled along the ray increases. Therefore, we introduce a caching mechanism to store the per-layer outputs of the previous tokens and only compute the attention of a new token in the current pass. This does not overcome the iterative loop during each forward pass but avoids redundant computations, which helps improve the decoding speed drastically when compared to the naive strategy. Due to computational constraints, we are only able to train GNT + AutoReg with much fewer rays sampled per iteration (500) when compared to other methods as discussed in Sec. 4.2. Tab. 4 discusses single scene optimization results on the LLFF dataset and we can clearly see that the GNT + AutoReg improves the overall performance when compared to existing baselines, and improving the PSNR scores in complex scenes (Orchids) when compared to our own method without the decoder. However, this is not consistent across all scenes and metrics. This could be because of the fewer number of rays sampled and we expect our results would improve when scaled to similar settings. Nevertheless, this shows that the learnable decoder predicts per-point RGB features effectively without any supervision from the volumetric rendering equation.

## C.2 ATTENTION GUIDED COARSE-TO-FINE SAMPLING

GNT's ray transformer learns point-to-point correspondences which helps model visibility and occlusion or more formally point-wise density $\sigma$. Motivated by this hypothesis, we estimate the depth maps from the extracted attention maps and qualitatively analyze the same in Sec. 4.5. Therefore, we could conclude that the learned point-wise importance values can be considered equivalent to point-wise density or $\sigma$. To further test our claim, we attempt to use the learned point-wise correspondence values to sample "fine" points, which are then queried to GNT to render a higher-quality image. Due to the set-like property of attention, we directly query the fine points to the same network without using a separately trained "fine" network unlike other NeRF methods (Mildenhall et al., 2020; Barron

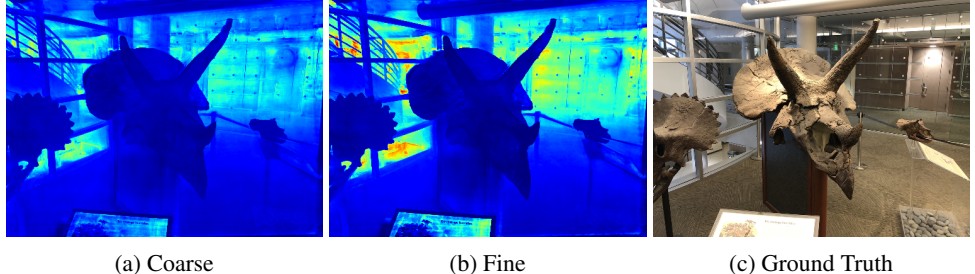

(a) Coarse         (b) Fine         (c) Ground Truth

Figure 9: Visualization of ray attention extracted during coarse, fine sampling where each color indicates the distance of each pixel relative to the viewing direction. The sampled fine points inferred from the learn attention values help GNT capture more fine-grained details. Red indicates far while blue indicates near.

et al., 2021; Wang et al., 2021b; Liu et al., 2022). Please note that we follow the same training strategy from Sec. 4.1 and only sample coarse-fine points during evaluation. Tab. 4 compares "GNT + Fine" against other methods, and we can clearly see that it outperforms other SOTA methods in complex scenes like Orchids, performing even better than our own method without fine sampling. However, the performance improvements are not significant across all scenes and we attribute this to the lack of training with the coarse-fine sampling strategy and expect our results to improve further. In Fig. 9, we visualize the estimated depth values obtained from the learned attention maps during both coarse, and fine stages. We can clearly see that the fine depth map is able to better estimate differences between nearby pixels which results in a higher resolution output.

## D  ADDITIONAL RESULTS AND ANALYSIS

**Breakdown of Table 1.**  Tables 6 and 7 include a breakdown of the quantitative results presented in the main paper into per-scene metrics. Our method quantitatively surpasses original NeRF and achieve on-par results with state-of-the-art methods. Although we slightly underperform NLF (Suhail et al., 2021) on some scenes, we argue that the comparison is not fair because NLF requires much larger batch size and longer iterations. We also include videos to demonstrate our results in the project page.

**Comparison with SRT (Sajjadi et al., 2022b).**  SRT (Sajjadi et al., 2022b) is another pure transformer based generalizable view synthesis baseline. In contrast to GNT, SRT barely utilizes attention blocks to interpolate views without any explicit geometry priors. We directly evaluate our cross-scene trained GNT in Sec. 4.3 on NMR dataset (Kato et al., 2018) without further tuning. In addition to SRT, we also include other generalizable novel view synthesis methods LFN (Sitzmann et al., 2021) and PixelNeRF (Yu et al., 2021) which are compared with SRT in Sajjadi et al. (2022b). All the results are presented in Tab. 5. Overall, we find GNT can largely outperform all baselines in all the metrics. We note that the pre-training data of SRT include the samples from NMR dataset (Kato et al., 2018), which is way more massive than GNT's pre-training datasets and has a narrower domain gap to the evaluation set. After all, our superior performance indicates our GNT can generalize better than SRT. We argue this might be because multi-view geometry is a strong inductive bias for novel view interpolation. That being said, a pixel on the novel view should be roughly consistent with its epipolar correspondence. Enforcing such constraints explicitly can significantly improve trainability and data efficiency. Nevertheless, we admit relieving multi-view geometry and learning a data-driven light transport prior from scratch can potentially render more sophisticated optics, which we leave for future exploration.

**Comparison with GPNR (Suhail et al., 2022).**  The concurrent work GPNR (Suhail et al., 2022) also utilizes fully attention-based architecture for neural rendering. Below, we summarize several key differences: **Embeddings:** GPNR leverages three forms of positional encoding (including light field embeddings) to encode the information of location, camera pose, view direction, etc. In contrast, GNT merely utilizes image features (with point coordinates). In this sense, GNT enjoys a neat design space and can potentially suggest such handcrafted feature engineering may not be necessary as

Table 5: Comparison with LFN, PixelNeRF, and SRT on the NMR (Kato et al., 2018) dataset.

| Models | PSNR↑ | SSIM↑ | LPIPS↓ | Avg↓ |
|---|---|---|---|---|
| LFN (Sitzmann et al., 2021) | 24.95 | 0.870 | - | - |
| PixelNeRF (Yu et al., 2021) | 26.80 | 0.910 | 0108 | 0.041 |
| SRT (Sajjadi et al., 2022b) | 27.87 | 0.912 | 0.066 | 0.032 |
| GNT | 32.12 | 0.970 | 0.032 | 0.015 |

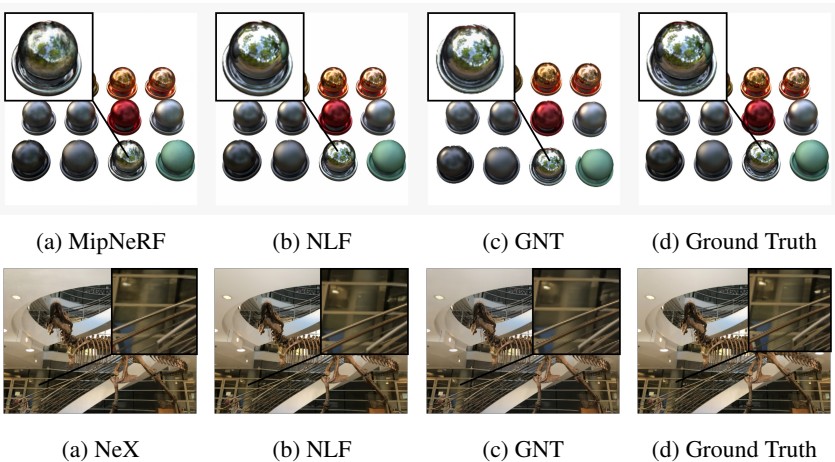

| (a) MipNeRF | (b) NLF | (c) GNT | (d) Ground Truth |
|---|---|---|---|

| (a) NeX | (b) NLF | (c) GNT | (d) Ground Truth |
|---|---|---|---|

Figure 10: Qualitative results for single-scene rendering. In the Trex scene from LLFF (first row) and Materials scene from Blender (second row), GNT's learnable renderer is able to model physical phenomenon like reflections.

they can be learned through cross-scene training. **Aggregation schemes:** GPNR has three-stage aggregation: 1) visual feature transformer to exchange information between patches across views, 2) epipolar aggregator transformer combines features along the epipolar line for each reference view, 3) reference view aggregator transformer fuses the epipolar information across multiple views. Such aggregation scheme extends NLF (Suhail et al., 2021) paper, which indicates GPNR rendering pipeline is more likely to simulate a light field-based rendering. Instead, GNT leverages the two-stage aggregations, which naturally correspond to the online scene representation and ray-based rendering in generalizable NeRF. In this sense, the rendering pipeline of GNT looks more like radiance field-based rendering. **Performance:** We have a direct comparison with GPNR on cross-scene generalization experiments (Tab. 2). Our results show that GNT can outperform GPNR on all the tested datasets with a simpler architecture and lighter feature engineering. Especially on the Shiny dataset (Tab. 2b), GNT significantly outperforms GPNR by ~3dB in PSNR.

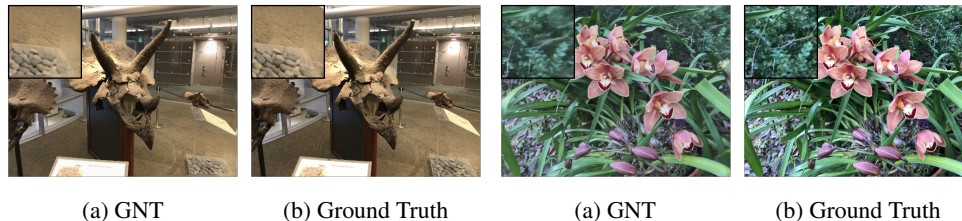

| (a) GNT | (b) Ground Truth | (a) GNT | (b) Ground Truth |
|---|---|---|---|

Figure 11: Qualitative comparison between images rendered by GNT and Ground truth image to discuss limitations. Epipolar correspondence for boundary pixels can be missing sometimes, which causes minor stripe artifacts.

Table 6: Comparison of GNT against SOTA methods for single scene rendering on the NeRF Synthetic Dataset (scene-wise).

| Models | Lego | Chair | Drums | Ficus | Hotdog | Materials | Mic | Ship |
|---|---|---|---|---|---|---|---|---|
| LLFF | 24.54 | 28.72 | 21.13 | 21.79 | 31.41 | 20.72 | 27.48 | 23.22 |
| NeRF | 32.54 | 33.00 | 25.01 | 30.13 | 36.18 | 29.62 | 32.91 | 28.65 |
| MipNeRF | 35.70 | 35.14 | 25.48 | 33.29 | 37.48 | 30.71 | 36.51 | 30.41 |
| NLF | 35.76 | 35.30 | 25.83 | 33.38 | 38.66 | 35.10 | 35.32 | 30.94 |
| GNT | 34.59 | 34.60 | 28.32 | 32.71 | 38.43 | 32.73 | 35.66 | 31.76 |

(a) PSNR↑

| Models | Lego | Chair | Drums | Ficus | Hotdog | Materials | Mic | Ship |
|---|---|---|---|---|---|---|---|---|
| LLFF | 0.911 | 0.948 | 0.890 | 0.896 | 0.965 | 0.890 | 0.964 | 0.823 |
| NeRF | 0.961 | 0.967 | 0.925 | 0.964 | 0.974 | 0.949 | 0.980 | 0.856 |
| MipNeRF | 0.978 | 0.981 | 0.932 | 0.980 | 0.982 | 0.959 | 0.991 | 0.882 |
| NLF | 0.989 | 0.989 | 0.955 | 0.987 | 0.993 | 0.990 | 0.992 | 0.952 |
| GNT | 0.984 | 0.986 | 0.966 | 0.986 | 0.989 | 0.984 | 0.993 | 0.906 |

(b) SSIM↑

| Models | Lego | Chair | Drums | Ficus | Hotdog | Materials | Mic | Ship |
|---|---|---|---|---|---|---|---|---|
| LLFF | 0.110 | 0.064 | 0.126 | 0.130 | 0.061 | 0.117 | 0.084 | 0.218 |
| NeRF | 0.050 | 0.046 | 0.091 | 0.044 | 0.121 | 0.063 | 0.028 | 0.206 |
| MipNeRF | 0.021 | 0.021 | 0.065 | 0.020 | 0.027 | 0.040 | 0.009 | 0.138 |
| NLF | 0.010 | 0.012 | 0.045 | 0.010 | 0.009 | 0.011 | 0.008 | 0.084 |
| GNT | 0.012 | 0.013 | 0.030 | 0.012 | 0.012 | 0.017 | 0.005 | 0.100 |

(c) LPIPS↓

| Models | Lego | Chair | Drums | Ficus | Hotdog | Materials | Mic | Ship |
|---|---|---|---|---|---|---|---|---|
| LLFF | 0.049 | 0.027 | 0.069 | 0.065 | 0.020 | 0.069 | 0.031 | 0.076 |
| NeRF | 0.018 | 0.016 | 0.043 | 0.020 | 0.017 | 0.025 | 0.013 | 0.047 |
| MipNeRF | 0.009 | 0.009 | 0.036 | 0.011 | 0.009 | 0.019 | 0.006 | 0.035 |
| NLF | 0.007 | 0.007 | 0.029 | 0.008 | 0.005 | 0.007 | 0.006 | 0.025 |
| GNT | 0.008 | 0.008 | 0.020 | 0.009 | 0.005 | 0.010 | 0.004 | 0.027 |

(d) Avg↓

Table 7: Comparison of GNT against SOTA methods for single scene rendering on the LLFF Dataset (scene-wise).

| Models | Room | Fern | Leaves | Fortress | Orchids | Flower | T-Rex | Horns |
|--------|------|------|--------|----------|---------|--------|-------|-------|
| LLFF | 24.54 | 28.72 | 21.13 | 21.79 | 18.52 | 20.72 | 27.48 | 23.22 |
| NeRF | 32.70 | 25.17 | 20.92 | 31.16 | 20.36 | 27.40 | 26.80 | 27.45 |
| NeX | 32.32 | 25.63 | 21.96 | 31.67 | 20.42 | 28.9 | 28.73 | 28.46 |
| NLF | 34.54 | 24.86 | 22.47 | 33.22 | 21.05 | 29.82 | 30.34 | 29.78 |
| GNT | 32.96 | 24.31 | 22.57 | 32.28 | 20.67 | 27.32 | 28.15 | 29.62 |

(a) PSNR↑

| Models | Room | Fern | Leaves | Fortress | Orchids | Flower | T-Rex | Horns |
|--------|------|------|--------|----------|---------|--------|-------|-------|
| LLFF | 0.932 | 0.753 | 0.697 | 0.872 | 0.588 | 0.844 | 0.857 | 0.840 |
| NeRF | 0.948 | 0.792 | 0.690 | 0.881 | 0.641 | 0.827 | 0.880 | 0.828 |
| NeX | 0.975 | 0.887 | 0.832 | 0.952 | 0.765 | 0.933 | 0.953 | 0.934 |
| NLF | 0.987 | 0.886 | 0.856 | 0.964 | 0.807 | 0.939 | 0.968 | 0.957 |
| GNT | 0.963 | 0.846 | 0.852 | 0.934 | 0.752 | 0.893 | 0.936 | 0.935 |

(b) SSIM↑

| Models | Room | Fern | Leaves | Fortress | Orchids | Flower | T-Rex | Horns |
|--------|------|------|--------|----------|---------|--------|-------|-------|
| LLFF | 0.155 | 0.247 | 0.216 | 0.173 | 0.313 | 0.174 | 0.222 | 0.193 |
| NeRF | 0.178 | 0.280 | 0.316 | 0.171 | 0.321 | 0.219 | 0.249 | 0.268 |
| NeX | 0.161 | 0.205 | 0.173 | 0.131 | 0.242 | 0.150 | 0.192 | 0.173 |
| NLF | 0.104 | 0.135 | 0.110 | 0.119 | 0.173 | 0.107 | 0.143 | 0.121 |
| GNT | 0.060 | 0.116 | 0.109 | 0.061 | 0.153 | 0.092 | 0.080 | 0.076 |

(c) LPIPS↓

| Models | Room | Fern | Leaves | Fortress | Orchids | Flower | T-Rex | Horns |
|--------|------|------|--------|----------|---------|--------|-------|-------|
| LLFF | 0.039 | 0.086 | 0.110 | 0.041 | 0.141 | 0.058 | 0.069 | 0.064 |
| NeRF | 0.028 | 0.073 | 0.112 | 0.036 | 0.121 | 0.055 | 0.056 | 0.058 |
| NeX | 0.025 | 0.057 | 0.077 | 0.027 | 0.102 | 0.037 | 0.038 | 0.040 |
| NLF | 0.016 | 0.053 | 0.062 | 0.022 | 0.084 | 0.030 | 0.029 | 0.030 |
| GNT | 0.017 | 0.055 | 0.061 | 0.021 | 0.087 | 0.042 | 0.031 | 0.028 |

(d) Avg↓

## E    DEFERRED DISCUSSION

**Discussion on Occlusion Awareness.**    Conceptually, view transformer attempts to find correspondence between the queried points and source views. The learned attention amounts to a likelihood score that a pixel on the source view is an image of the same point in the 3D space, i.e., no points lies between the target point and the pixel. NeuRay (Liu et al., 2022) leverages the cost volume from MVSNet (Yao et al., 2018) to predict per-pixel visibility and shows that introducing occlusion information is beneficial for multi-view aggregation in generalizable NeRF. We argue that instead of explicitly regressing the visibility, purely relying on epipolar geometry-constrained attention can automatically learn how to infer occlusion, given prior works in Multi-View Stereo (MVS) (Yang et al., 2022; Ding et al., 2021). In view transformer, the U-Net provides multi-scale features to the transformer, and the attention block acts as a matching algorithm, which selects the pixels from neighboring views that maximize view consistency. We defer empirical discussion to Sec. 4.5.

**Discussion on Depth Cuing.**    The ray transformer iteratively aggregates features according to the attention value. This attention value can be regarded as the importance of each point to form the image, which reflects visibility and occlusion reasoned by point-to-point interaction. Therefore, we can interpret the average attention score for each point as the accumulated weights in volume rendering. In this sense, we can infer the depth map from the attention map by averaging the marching distance $t_i$ with the attention value. This implies our ray transformer learns geometry-aware 3D semantics on both feature space and attention map, which helps it generalize well across scenes. We defer visualization and analysis to Sec. 4.5. NLF (Suhail et al., 2021) proposes a similar two-stage rendering transformer, but it first extracts features on the epipolar lines and then aggregates epipolar features to get the pixel color. We doubt this strategy may fail to generalize as epipolar features lack communication with each other and thus cannot induce geometry-grounded semantics.

## F    LIMITATIONS

Although our method achieves strong single-scene performance and achieves SOTA cross-scene generalization, we discuss certain limitations of our method. The view transformer relies on epipolar constraints so that it can only aggregate information from valid epipolar lines. Therefore, non-epipolar scenes and complex light transport might not be captured by the view transformer. Although we adopt a feature extractor with large receptive fields to encode global light transport and our view transformer empirically works well on complex lighting effects, what is captured by the image encoder remains unclear. Moreover, epipolar correspondence for the boundary pixels sometimes are missing, which causes some minor artifacts (see Fig. 11).

