# OpenReview forum: "Is Attention All That NeRF Needs?"
_ICLR.cc/2023/Conference — ICLR 2023 poster_

### Official Review · Reviewer_xoKn · 2022-10-20

**Confidence:** 4
**Correctness:** 4
**Technical Novelty And Significance:** 3
**Empirical Novelty And Significance:** 3
**Recommendation:** 8

**Clarity, Quality, Novelty And Reproducibility:**

The paper is well-written with good motivation and clear descriptions of methods and experiments.
Details are included in the implementation subsection, with a good chance for the method to be reproduced.


**Strength And Weaknesses:**

Strength
1. View transformer enables the learning of a neural representation generalized across scenes.
2. Replacing explicit volume rendering formula with a learned ray transformer. This might contribute to better render refractions and reflection.
3. Outperforming other state-of-the-art methods in both single-scene and cross-scene scenarios.

Weakness
1. GNT includes two transformers. It is not clear how stable is the training with respect to random seed or changing some hyperparameters.
2. Some clear artifact at the bottom parts in the Horns example.

**Summary Of The Paper:**

The paper introduces a novel approach based on two transformers to reconstruct NeRF. The view transformer leverages multi-view geometry (epipolar geometry) to guide the attention-base scene representation. Next, the ray transformer renders novel views using attention to decode from the representation learned by the view transformer. The proposed GNT outperforms other state-of-the-art methods in both single-scene and cross-scene scenarios.

**Summary Of The Review:**

This is one of the first papers using the attention mechanism in most components for the task of NeRF.
Two novel transformers are introduced to learn a neural representation generalized across scenes and better render refractions and reflection. Finally, The proposed GNT outperforms other state-of-the-art methods in single-scene and cross-scene scenarios.

---

> ### Author Response · Authors · 2022-11-16
> **Response to Reviewer xoKn**
>
> Dear Reviewer xoKn,
>
> We thank the reviewer xoKn for appreciating our works’ novelty and empirical significance. We summarize our point-to-point responses as below.
>
> **Q1. It is not clear how stable is the training with respect to random seed or changing some hyperparameters.**
>
> We observe the training of GNT is quite stable, although changing seed may have ~0.1dB PSNR deviation. For hyperparameter choice, we find larger batch size, more sampled points per ray, and longer training time always boost the performance. We do not carefully tune the hyperparameters and training recipe. Our latest results are produced by 2048 rays/batch, 192 points/ray, and 500k training steps, which maximally fits our GPU capacity. We will release all the code and training hyperparameters upon acceptance.
>
> **Q2. Some clear artifact at the bottom parts in the Horns example.**
>
> Thanks for this nice catch. The mild artifacts around the borders of the rendered image are caused by the invalid epipolar correspondence. This is an inherent limitation of our proposed method which we have included in our newly added limitation section in Appendix F. A practical way to get rid of this artifact is to do central cropping of the target view.
>
> Paper #3027 Authors

---

> > ### Comment · Reviewer_xoKn · 2022-12-02
> > **Response to rebuttal**
> >
> > My concerns are addressed in the rebuttal. I also read other reviewers and appreciated other reviewers' comments.
> > Ultimately, I still think this work shows enough novelty as it is the first pure transformer-based NeRFs with reasonably good performance.

---

> > > ### Author Response · Authors · 2022-12-02
> > > **Thanks**
> > >
> > > Dear Reviewer xoKn,
> > >
> > > Thank you for your effort on reviewing our paper and keeping endorsing our work. We would be more than glad to resolve any further concerns.
> > >
> > > Paper #3027 Authors

---

### Official Review · Reviewer_TeHV · 2022-10-24

**Confidence:** 4
**Correctness:** 3
**Technical Novelty And Significance:** 3
**Empirical Novelty And Significance:** Not applicable
**Recommendation:** 6

**Clarity, Quality, Novelty And Reproducibility:**

Paper writing is clear, and the design of view transformer and ray transformer are novel.

**Strength And Weaknesses:**

Strength: This is, to my knowledge, the first transformer based NeRF reconstruction method. While the proposed method is not groundbreaking, it is a reasonable design and an advancement to study how NeRF reconstruction can be achieved by transformer.

Weakness: I would expect more ablation study and comparisons of the proposed GNT with some more recent methods for NeRF reconstruction not limited to the vanilla NeRF and the MipNeRF. Indeed, there are many recent advancement in NeRF for speeding up the reconstruction, or use partial depth, or spherical harmonic coefficients for NeRF reconstruction. These methods are not referenced nor compared in the paper. Such addition references, comparisons and ablation study should be able to further strengthen the contribution of this work. In the current submission, this paper just read like this is another alternative for NeRF reconstruction, but the advantages of GNT over other recent NeRF reconstruction methods are not obvious.

**Summary Of The Paper:**

This paper presents the first NeRF reconstruction method, Generalizable NeRF Transformer (GNT), based on transformer. It introduces the view transformer which predicts coordinate-aligned features by aggregating information from epipolar lines on the neighboring views with attention and the ray transformer which renders novel views using attention to decode the features from the view transformer along the sampled points. Experimental results show that the GNT can successful reconstruct a NeRF model with more details.

**Summary Of The Review:**

I am happy with this submission and I think the technical novelty of this work is sufficient to be accepted in ICLR. At the same time, I feel that this paper can be even better with further ablation study and comparisons (check my comments in weakness).

---

> ### Author Response · Authors · 2022-11-16
> **Response to Reviewer TeHV**
>
> Dear Reviewer TeHV,
>
> We thank the reviewer TeHV for giving us a positive initial assessment and constructive suggestions. For your questions on comparison with more baselines, please see our responses below.
>
> **Q1. Comparisons of the proposed GNT with some more recent methods for NeRF reconstruction not limited to the vanilla NeRF and the MipNeRF.**
>
> Thanks for pointing out these recent advancements in NeRF. To our understanding, speed-up NeRFs are orthogonal to our work. But we note speedup GNT can be a durable future direction because the pure transformer architecture of GNT is friendly for hardware acceleration on edge devices [1]. In this paper, we mainly discuss the standard setting of NeRF training and we initially do not involve extra information e.g., partial depth. Spherical harmonic coefficients are proposed for voxel-based NeRF, and to our best knowledge, they should have comparable performance with the original NeRF. Per reviewer’s request, we provide the table below for an intuitive comparison (on Blender dataset) with representative works towards the directions mentioned by the reviewer.
>
> | Method | PSNR | SSIM | LPIPS |
> | --- | --- | --- | --- |
> | E-NeRF [2] | 26.55 | 0.942 | 0.088 |
> | Plenoxel [3] | 31.71 | 0.954 | 0.049 |
> | Plenoctree [4] | 31.71 | 0.958 | 0.066 |
> | GNT | 33.71 | 0.975 | 0.025 |
>
> [1] Li et al. EfficientFormer: Vision Transformers at MobileNet Speed.
>
> [2] Lin et al. Efficient Neural Radiance Fields with Learned Depth-Guided Sampling
>
> [3] Yu et al. Plenoxels Radiance Fields without Neural Networks
>
> [4] Yu et al. PlenOctrees For Real-time Rendering of Neural Radiance Fields
>
>
>
> **Q2. This paper just read like this is another alternative for NeRF reconstruction, but the advantages of GNT over other recent NeRF reconstruction methods are not obvious.**
>
> The most significant advantage of GNT over NeRF is that GNT can be pre-trained across multiple scenes and achieve zero-shot generalization to unseen scenes. This avoids the tedious per-scene optimization for new scenes in classical NeRF. Compared with all existing generalizable NeRF, GNT achieves the best result (see Tab. 2) because it “neuralizes” the rendering pipeline and can learn a more generalizable rendering function from observed data. Moreover, with a learned renderer, GNT can better fit challenging scenes with complex lighting transports (see Fig. 4 and newly added Tab. 2).
>
> Paper #3027 Authors

---

### Official Review · Reviewer_qn8f · 2022-10-25

**Confidence:** 4
**Correctness:** 3
**Technical Novelty And Significance:** 2
**Empirical Novelty And Significance:** 2
**Recommendation:** 6

**Clarity, Quality, Novelty And Reproducibility:**

The overall writing is Okay. Figure 2 is confusing because of lack of explanation. The originality of the paper is limited.

**Strength And Weaknesses:**

+ The paper demonstrates that pure transformer architecture with geometry prior can learn to perform the task of novel view synthesis.
+ The model has a better performance compared with other baselines in the cross-scene generalization settings.

- My major concern about the paper is the lack of technical novelty compared with previous works. There are a number of works that leverage transformers to learn the pipeline of volume / light field rendering such as [1][2][3][4]. Specifically, NeRFormer [1] also uses the transformer for aggregating features of every 3d point from feature maps of multi-view images using epipolar constraints. GNT simply replaces the volumetric rendering part with a transformer, which introduces limited novelty.
- The benefits of the proposed architecture is not empirically convincing. GNT performs worse than NLF[4] in the single-scene settings. On cross-scene settings, the improvement is also very marginal. I would like to know the computing cost (training time inference time) of GNT compared with GNT with volumetric rendering ablation.
- The authors claim that the GNT can better handle challenging view-dependent effects. However, no quantitative results (e.g. on the Shiny dataset) can support this point. And in Figure 4, results of the strongest baseline NLF is not shown. This makes it hard to evaluate the effectiveness of GNT for view-dependent effects.

[1] Common Objects in 3D: Large-Scale Learning and Evaluation of Real-life 3D Category Reconstruction. ICCV 2021.
[2] Scene Representation Transformer: Geometry-Free Novel View Synthesis Through Set-Latent Scene Representations. CVPR 2022.
[3] IBRNet: Learning Multi-View Image-Based Rendering. CVPR 2021.
[4] Light Field Neural Rendering. CVPR 2022.

A few questions:
1. In Table 3, why the Epipolar Agg. → View Agg variant is significantly worse than NLF when they share a similar two-stage aggregation strategy?
2. What is the training/inference time of GNT in single-scene and cross-scene settings?

Minor:
- Section 4.3 Datasets section, “IBRNetto” -> “IBRNet to”
- In Figure 2, symbols of input should have explanations. Otherwise it is very hard to understand the figure.


**Summary Of The Paper:**

The paper proposes a neural architecture GNT that uses transformers for novel view synthesis from multi-view images. Specifically, a view transformer aggregates feature from feature maps of input images to obtain features of sampled 3D points in the scene. Then a ray transformer learns to aggregate features of points along each ray and decode code from them. Experiments have shown that GNT can achieve comparable results for single-scene overfitting settings. And achieve better results on cross-scene generalization settings compared with other baselines.

**Summary Of The Review:**

Overall, the submission presents marginal improvements over existing method. However, the main concern is the lack of technical contribution, and that the empirical evaluation is not convincing.

---------
UPDATED: after rebuttal I think my concerns have been partly addressed. I changed my score to weak accept.

---

> ### Author Response · Authors · 2022-11-16
> **Response to Reviewer qn8f**
>
> Dear Reviewer qn8f,
>
> We thank the reviewer qn8f for reviewing our paper and providing meaningful comments. The main concern of reviewer qn8f lies in our originality and empirical soundness. Regarding originality, we hope to highlight that our paper is the first work using a transformer to universally learn a rendering function, which is significant and goes beyond all the mentioned transformer-based NeRF. Below we elaborate on our point-by-point responses.
>
> **Q1.   Lack of technical novelty compared with previous works.**
>
> We are familar with existing methods combining transformers with NeRF, and we have discussed them in Sec. 2. As kindly accredited by Reviewer TeHV, our GNT is the first pure transformer-based NeRFs, and our major innovation that has not been done in prior works and is new to GNT is that it replaces the volumetric rendering part with a transformer. This is not a “simply” random change, but rather supported by thoughtful motivation and convincing empirical results.
>
> First of all, the volume rendering equation adopted in NeRF over-simplifies the optical modeling of solid surface, reflectance, inter-surface scattering and other effects. . Technically, the full formulation of volume rendering (radiative transfer equation, as used in modern volume path tracers) is capable of handling all these effects. However, NeRF uses a simplified formulation which does not simulate all physical effects. This implies that radiance fields along with volume rendering in NeRF are not a universal imaging model, which may have limited the generalization ability of NeRFs as well. And when it is adopted in generalizable NeRF (e.g., IBRNet, NeRFormer), it may further restrict the generalization capability. This motivates us to propose a data-driven rendering function. Our key innovation is to use a transformer to model a black-box renderer and our insight is that a ray-based rendering can be regarded as a sequence function while a transformer can universally model any sequence functions [1][2]. That being said, the learned ray transformer can go beyond volume rendering. Combined with the view transformer, GNT attains a fully neuralized and data-driven pipeline which unleashes the generalization capability of the NeRF and eliminates the inductive bias of scene modelling and rendering functions. In Tab. 2, we have added quantitative results of pre-trained GNT when generalizing to the shiny dataset which has complex optics. We also updated Fig. 4 with our generalization results. Those numbers and visualizations demonstrate the superior cross-scene generalization performance of GNT, which is even better than many per-scene optimized NeRF and on par with the per-scene training SOTA NLF.
>
> As we have thoroughly and faithfully discussed in the paper, GNT indeed stands on the shoulders of several giants, but with critical differences from them, besides replacing volumetric rendering with a ray transformer. For example, both IBR-Net and NeRFormer decode the latent feature representation to point-wise color and density, and rely on classic volume rendering to form the image, while we render the target pixel from latent features directly - and more flexibly, thanks to our learnable rendering by ray transformer. NLF also proposes a transformer to simulate light field rendering but their rendering scheme was not generalizable across scenes thus requires per-scene fitting.
>
>
> [1] Yun et al. Are transformers universal approximators of sequence-to-sequence functions? ICML 2019
>
> [2] Lee et al. Set Transformer: A Framework for Attention-based Permutation-Invariant Neural Networks. ICML 2019

---

> > ### Author Response · Authors · 2022-11-16
> > **Response to Reviewer qn8f (cont.)**
> >
> > **Q2. The benefits of the proposed architecture are not empirically convincing.**
> >
> > We want to note that on single-scene experiments, NLF uses a much heavier training setting with 500,000 iterations and 16,384 batch size. Although their batch size remains prohibitive for us, we train our GNT with the same iterations (with smaller batch size of 2048), and the updated results are reported in Tab 1 in our revision. Accordingly, our GNT can already perform as well as NLF in Blender and LLFF dataset.  We would like to emphasize that GNT takes advantage over NLF in its generalization ability. **In our newly added Tab. 2, a pre-trained GNT can even achieve comparable results with per-scene optimized SOTA NLF on shiny datasets without further tuning.**
> >
> > For cross-scene experiments, our improvement is not marginal. GNT can outperform all other methods by ~10% in the Avg. metric. Moreover, GNT improves previous SOTA IBRNet by ~0.5dB in PSNR. We also note that NeuRay requires additional depth maps, and is pre-trained on an extra DTU dataset. GNT can even surpass it in Avg. metric without any of those auxiliary information. **Moreover, we added generalization results on shiny datasets in Tab. 2. Our GNT outperforms the SOTA GPNR and even many per-scene optimized NeRF.**
> >
> > **Q3. No quantitative results (e.g. on the Shiny dataset) can support this point. And in Figure 4, results of the strongest baseline NLF is not shown.**
> >
> > We thank reviewer qn8f for pointing this out. Given the limited rebuttal period, we are not able to finish per-scene training on every scene in the shiny dataset. However, we indeed evaluated our pre-trained GNT on the shiny dataset, and updated Fig. 4 to demonstrate our generalization results and compare with NLF. Our pre-trained GNT can attain comparable results with NLF. This performance is notable because GNT without any per-scene optimization can even achieve on-par results with the SOTA per-scene training methods.
> >
> > **Q4. Why the Epipolar Agg. → View Agg variant is significantly worse than NLF when they share a similar two-stage aggregation strategy?**
> >
> > Although “Epipolar Agg. -> View Agg.” should conceptually resemble NLF, the implementation details and training recipe differ from NLF. For example, NLF prepends a light field encoding as an extra token and includes multiple embeddings while we barely use images features; NLF uses GAT-like attention to weighted average the whole epipolar line while we simply take an average over the whole sequence. The purpose of this ablation study is to support the designs of GNT. Our finding is that NLF-like aggregation may not achieve comparable results without careful modification on the input space and architectures.
> >
> > **Q5. Regarding training/inference time.**
> >
> > Thanks for pointing this out. During training, GNT requires a total of 0.3 seconds for one iteration (including epipolar projection)  with 2048 rays in one batch, and 192 points sampled in each ray. During inference, GNT takes 0.03 seconds for a similar iteration. GNT processes input rays faster as it discards the additional volumetric rendering step and rather directly predicts the target pixel color.
> >
> > **Q6. In Figure 2, symbols of input should have explanations.**
> >
> > Thanks for this suggestion. We have added explanations of inputs for Fig. 2 in our revision.
> >
> > Paper #3027 Authors

---

### Official Review · Reviewer_i8zE · 2022-11-04

**Confidence:** 4
**Clarity, Quality, Novelty And Reproducibility:** Please, refer to the strengths above.
**Correctness:** 4
**Technical Novelty And Significance:** 3
**Empirical Novelty And Significance:** 4
**Recommendation:** 8

**Details Of Ethics Concerns:**

No concerning ethical issue as far as can be seen.

**Strength And Weaknesses:**

++ The addressed problem in this paper is a problem of high interest.

++ The proposed method is intuitive and meaningful.

++ The paper is well written and easy to follow.

++ The provided experimental results are comparable, and in fact exciting.

++ The experimental evaluations provided in the supplementary material further illustrate the effectiveness of the proposed method.

-- I do not find any major problem with the proposed method. However, a discussion regarding the failure cases and limitations is missing. The paper can benefit from demonstrating the examples where the proposed method lacks to generalize.

**Summary Of The Paper:**

 This paper addresses the problem of generalizable Nerf using a view transformer for generalization and a ray transformer for ray-based rendering.  For generalization, the view transformer uses multi-view geometry inductive bias to predict coordinate aligned features by aggregating information along the epipolar lines on other views. For rendering, the ray transformer renders novel views using features from the view transformer along the ray points during ray marching. Such designs allow authors to generalize the Nerf in the previously unseen scenes.

**Summary Of The Review:**

Please, refer to the weakness regarding the failure cases and limitation of the proposed method.

---

> ### Author Response · Authors · 2022-11-16
> **Response to Reviewer i8zE**
>
> Dear Reviewer i8zE,
>
> We sincerely thank the reviewer i8zE for advocating our novelty, soundness, significance, as well as all the valuable comments and constructive suggestions. Regarding your suggestions on mentioning limitations and failure cases, see our responses as below.
>
> **Q1. Discussion regarding the failure cases and limitations is missing.**
>
> Our view transformer relies on epipolar constraints, which can only aggregate information from the corresponding epipolar lines. Non-epipolar scenes and complex light transport might not be captured by the view transformer. Although we adopt a feature extractor with large receptive fields to encode global light transport and our view transformer empirically works well on complex lighting effects, what is captured by the image encoder remains unclear. Moreover, epipolar correspondence for the boundary pixels sometimes are missing, which causes some minor artifacts. We add a corresponding section in Appendix. F to discuss limitations and illustrative examples in Fig. 11.
>
> A minor reminder: we notice reviewer i8zE kindly acknowledged the soundness of our method with the sentences such as “I do not find any major problem with the proposed method.”, but gave us rate 1 in correctness. We suspect this was a mistake and would sincerely appreciate it if reviewer i8zE could double-check.
>
> Paper #3027 Authors

---

> > ### Comment · Reviewer_i8zE · 2022-12-02
> > **Response to rebuttal**
> >
> > I read all reviews and authors' reply. I found out that I was not fully aware of the existing works. The raised concerns and provided discussion are relevant. In this regard, authors' provided reply is largely convincing to me, which I think is necessary to be added in the camera-ready. I can foresee how the the camera-ready going to look like.Therefore, I would like to keep my original rating. Also, thank you for raising "a minor reminder" regarding my fortunate mistake about correctness. This has been amended.

---

> > > ### Author Response · Authors · 2022-12-02
> > > **Thanks**
> > >
> > > Dear Reviewer i8zE,
> > >
> > > Thank you again for advocating our paper and providing thoughtful comments on adding a limitation section. We have included one in our Appendix F, which should be also presented in our camera-ready version. Please kindly let us know any concerns we need to resolve in our camera ready. We will make necessary change and all that.
> > >
> > > Paper #3027 Authors

---

### Author Response · Authors · 2022-11-18
**Overall Response**

We thank all reviewers for their thoughtful comments and feedback. The reviewers agree on that our paper is novel (TeHV, xoKn), well-written (i8zE, TeHV, xoKn), and includes extensive experiments to support our proposal (i8zE, xoKn).  We have addressed all the reviewers’ questions in our responses. Here we highlight that our proposed GNT is one of the first pure transformer based NeRF that also achieves significant generalization results. Both scene representation and differentiable rendering equation are learned by transformers. Below we summarize some key changes we made to improve the quality of our paper per reviewers’ requests.

- Update Tables. 1, and 2(a) with newer results after training GNT for as many iterations as compared methods.
- Include quantitative results on the shiny dataset (Table. 2(b)) to evaluate GNT’s generalization capability. GNT manages to outperform all SOTA methods for cross-scene generalization and even achieve comparable results with per-scene optimized NLF, without further tuning (discussed in Sec. 4.3).
- Discuss limitations in Appendix. F with illustrative examples in Fig. 11.

Again, we thank all reviewers for their time and effort reviewing this paper and providing constructive suggestions to  improve our work.

Paper #3207 Authors

---

### Decision · Program_Chairs · 2023-01-20

**Decision:**

Accept: poster

**Justification For Why Not Higher Score:**

Because the paper though technically sound has very close related works, e.g., Sugail et al. or Nerformer that have very similar architectures and motivations.

**Justification For Why Not Lower Score:**

The paper proposes a method for view prediction that generalizes across scenes using two transformer modules, one for scene encoding by attention along epipolar lines and another for ray-based rendering. All the reviewers agree on the good empirical performance of the method. Reviewers raised concerns regarding comparing to additional baselines, as well as SRT and Suhail et al. 2022 which the rebuttal successfully addressed.

**Metareview: Summary, Strengths And Weaknesses:**

The paper proposes a method for view prediction that generalizes across scenes using two transformer modules, one for scene encoding by attention along epipolar lines and another for ray-based rendering. All the reviewers agree on the good empirical performance of the method. Reviewers raised concerns regarding comparing to additional baselines, as well as SRT and Suhail et al. 2022 which the rebuttal successfully addressed.

**Note From Pc:**

if the above contains the word "oral" or "spotlight" please see: "oral" presentation means -> notable-top-5% and "spotlight" means -> notable-top-25%. As stated in our emails, we are disassociating presentation type from AC recommendations

**Summary Of Ac-Reviewer Meeting:**

N/A